# MFS: A Saliency Driven Interactive Multimodal Fusion Framework for Robust Semantic Segmentation in Complex and Occluded Scenes

## Abstract

In complex scenes, semantic segmentation often encounters challenges such as difficulty in detecting distant small or weak targets and recognizing occluded objects. Existing methods still suffer from limited robustness and suboptimal multimodal feature fusion. To address these issues, this paper proposes an interactive multimodal semantic segmentation framework based on frequency domain dynamic routing and activation region guidance, which effectively enhances the feature extraction capability, fusion robustness, and semantic representation of multimodal images. The proposed framework consists of three core modules: first, an edge feature enhancement module that performs fine-grained selection of key regions on the initial features to enhance weak targets and edge details; second, an activation region guided hybrid attention module that effectively fuses prominent region information from infrared and visible modalities; and finally, a deep semantic enhancement learning module that incorporates dynamic convolutional masks to improve the semantic consistency of fused features at both global and local levels. Experimental results on multiple public datasets demonstrate that the proposed method outperforms existing approaches in terms of image fusion quality, segmentation accuracy, and object detection performance, showing especially strong robustness and generalization ability in complex and occluded scenes.

## 1 Introduction

Semantic segmentation, as a core technology for pixel-level scene understanding, plays a vital role in areas such as autonomous driving and medical image analysis. In autonomous driving, it enables accurate recognition of roads, obstacles, and pedestrians Seichter et al. (2021); Wu et al. (2025b), while in the medical domain, it facilitates precise localization and analysis of lesions Hao et al. (2024); Zhang et al. (2025b). However, current semantic segmentation methods face two major challenges, as shown in Figure 1: they often struggle to detect small or low-signal (weak) targets at long distances, and they have difficulty perceiving partially occluded objects. To enhance model robustness in complex scenarios, multimodal image fusion methods have attracted increasing attention—particularly infrared and visible image fusion—which has shown significant advantages in military reconnaissance and nighttime surveillance applications Li et al. (2018); Lu et al. (2020).

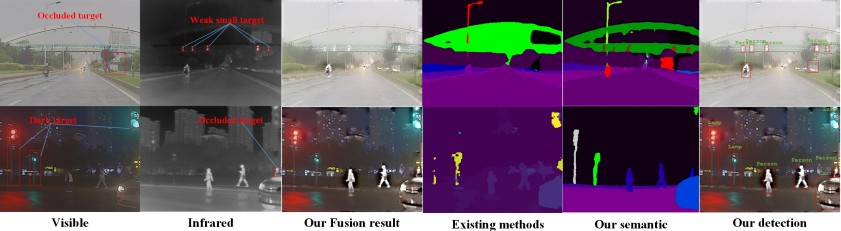

**Figure 1:** This paper proposes a multimodal fusion framework for weak and occluded targets in complex scenes, enabling accurate object detection and segmentation.

Although deep learning has driven rapid progress in image fusion technologies in recent years—with methods based on autoencoders Li & Wu (2019), convolutional neural networks Ma et al. (2021), and generative adversarial networks Ma et al. (2019; 2020)—existing approaches still face two fundamental technical issues. First, most current methods lack a unified cross-modal representation mechanism. Due to the significant heterogeneity between infrared and visible images in terms of imaging principles, semantic structures, and texture details, existing approaches often rely on simple feature concatenation or alignment strategies, which fail to deeply model the shared and complementary features across modalities Geng et al. (2024). Second, most fusion frameworks use fixed or heuristic fusion rules, lacking the ability to dynamically adapt to different scene conditions. As a result, their generalization performance and robustness in real-world applications remain limited Liu et al. (2022).

To address the above issues, this paper proposes an interactive multimodal semantic segmentation framework based on frequency domain dynamic routing and activation region guidance. The framework enhances weak targets and edge details in images by leveraging frequency energy path selection and interactions between high and low frequency components. Additionally, a hybrid attention module guided by activation regions is introduced to adaptively focus on high quality features, enabling precise fusion of complementary information from infrared and visible modalities. Finally, a deep semantic mask learning strategy, combining global and local features, is introduced to improve the semantic consistency and discriminability of the fused features, thereby significantly enhancing segmentation performance and robustness. This method systematically improves multimodal feature extraction, information fusion, and semantic understanding, significantly boosting the visual quality of fused images, semantic segmentation accuracy, and object detection performance. The three main innovations of this paper are as follows:

In summary, (1) For multimodal fusion, a method combining dynamic frequency-domain energy and activation region-guided attention is proposed to enhance feature robustness and achieve precise multimodal fusion. (2) For semantic segmentation, a hierarchical semantic learning approach is introduced, which captures deep semantic information based on dynamic masks of global and local regions. (3) The proposed multimodal fusion framework excels in semantic segmentation, image fusion quality, and object recognition.

## 2 RELATED WORK

Infrared–visible image fusion is vital for semantic segmentation. Current methods mainly include feature-level, attention-based, and deep interactive fusion to enhance accuracy and robustness.

### 2.1 FEATURE LEVEL FUSION METHODS

Early multimodal research primarily employed encoder-decoder architectures for feature fusion. FuseNet Hazirbas et al. (2016) pioneered multimodal fusion for semantic segmentation, but simple feature concatenation or weighting struggled to deeply model cross-modal correlations. Ferrod et al.'s CroDiNo-KD Ferrod et al. (2025) improved modality alignment through disentangled distillation; however, distillation of shallow features limited deep interaction and caused information loss. Chen et al.'s TransUNet Chen et al. (2021) leveraged Transformers to enhance single-modality representation in medical imaging but lacked sufficient cross-modal interaction. Wei et al. Wei et al. (2023) pointed out that shallow fusion in nighttime segmentation failed to capture deep illumination information. Overall, shallow fusion provides limited feature information and easily loses complementary information, restricting support for semantic segmentation. To address this, this paper proposes a frequency-domain energy-driven dynamic routing method to improve the robustness of bimodal features, and incorporates frequency features for interactive modeling, thereby providing rich information for subsequent fusion.

### 2.2 ATTENTION FUSION METHOD

Attention mechanisms are important for salient regions in images. Zhang et al. Zhang et al. (2021) introduced RFN-Nest, which combined channel and spatial attention modules. Chen et al. Chen et al. (2022) proposed RegionViT, which integrates regional and local attention mechanisms to capture the global contextual information required for multimodal fusion. Yu et al. Qi et al. (2025)

developed a cross-modality enhancement module that models both intra- and inter-modality dependencies through cross-modality attention, thereby improving the feature fusion capability between infrared and visible modalities. Although attention mechanisms have played a crucial role in multimodal fusion, two major limitations still lead to suboptimal fusion performance: an over-reliance on global average pooling in the attention mechanism may cause the loss of local details such as edge textures; and static attention weights cannot dynamically adapt to scene-related changes in feature distributions. To address these issues, this paper proposes an activation region guided fusion method. Instead of directly fusing features through attention, the method first focuses on activation regions and then selectively guides attention features for precise fusion. This approach can dynamically adapt to scene-related changes in feature distributions.

## 2.3 DEEP INTERACTIVE FUSION METHODS

With the rise of the Transformer architecture, researchers have begun to explore deeper cross-modal interaction mechanisms. Chen et al. Li et al. (2024a) proposed a cross-modal network based on the Swin Transformer, utilizing hierarchical cross-attention to achieve feature reorganization. Liu et al. Liu et al. (2023b) introduced a hybrid network incorporating deformable convolutions, which effectively enhances the model's ability to capture semantic information from complex visual features. Kim et al. Kim et al. (2024) presented a novel graph-structured modeling network that performs well in complex urban scenes. In the latest research, Jiang et al. Jiang & Shen (2024) proposed a Swin Transformer based cross modal network to enhance medical image fusion. Although the above methods have achieved significant performance improvements, they still incur high computational costs Chaudhary et al. (2024); Yuan et al. (2024); Zhao et al. (2024a). Moreover, these methods' heavy reliance on Transformer architectures often results in insufficient modeling and perception of local image semantic details, limiting their ability to learn semantic information of edge regions as well as small and weak targets. To this end, this paper proposes a module that integrates Transformer and masked convolutional filtering to achieve joint perception of local and global semantics. Meanwhile, by adopting the Transformer optimization strategies from Shen et al. (2021), the model significantly improves computational efficiency while maintaining segmentation accuracy.

## 3 METHOD

Figure 2 shows the overall framework of this paper, which consists of three modules: (1) A dynamic frequency domain feature enhancement module that addresses issues such as the lack of detail in infrared images, high noise in visible images, and the difficulty of simultaneously extracting complete weak target features from both modalities; (2) A activation region guided fusion enhancement modules designed to avoid occlusion neglect commonly found in naive fusion approaches; and (3) A hierarchical semantic feature enhancement module is dedicated to improving the high level semantic representation ability in segmentation and detection tasks.

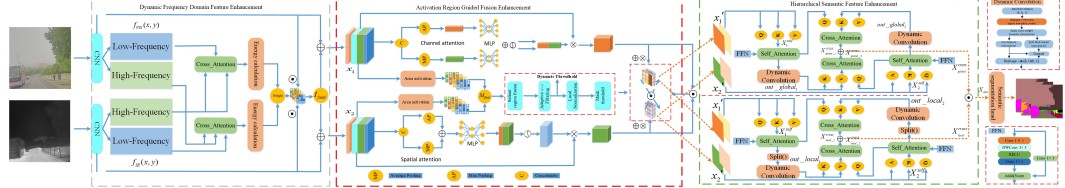

**Figure 2:** A saliency driven interactive multimodal fusion framework for robust semantic segmentation in complex occluded scenes, with three modules enhancing feature representation, fusion, and semantic alignment.

## 3.1 DYNAMIC FREQUENCY DOMAIN FEATURE ENHANCEMENT

Infrared and visible images differ significantly in imaging principles and information representation, and existing methods often fail to comprehensively extract cross-modal information. To address this issue, as shown in Figure 3 (which clearly illustrates that the energy distribution across frequency-domain regions varies significantly between modalities, and only the frequency-domain energy corresponding to the target regions can provide richer information), we propose a dynamic

routing module based on frequency-domain energy computation. This module automatically selects the frequency fusion path according to the frequency-domain energy and its corresponding regions.

Our module adopts a novel frequency energy selection method to enhance multimodal feature extraction by integrating frequency domain decomposition and energy guided routing. For spatial domain features extracted by CNN $f_{\text{IR}}(x, y)$ and $f_{\text{VIS}}(x, y)$, we apply $2D$ FFT to obtain frequency domain representations: $F_{\text{IR}}(u, v) = \mathcal{F}(f_{\text{IR}}(x, y))$, $F_{\text{VIS}}(u, v) = \mathcal{F}(f_{\text{VIS}}(x, y))$. where $\mathcal{F}(\cdot)$ encodes magnitude and phase. To separate frequency bands, we design two masks: a low-pass filter mask $M_{low}(u, v)$ for contours and smooth areas, and a high-pass filter mask $M_{high}(u, v) = 1 - M_{low}(u, v)$ for edges and textures. These masks are element-wise multiplied with the frequency features for decomposition:

$$F_{IR}^{i} = F_{IR}(u, v) \cdot M_i(u, v), i \in (low, high)$$
$$F_{VIS}^{i} = F_{VIS}(u, v) \cdot M_i(u, v), i \in (low, high)$$

$$(1)$$

where $F_{IR}^{low}, F_{IR}^{high}, F_{VIS}^{low}, F_{VIS}^{high}$ represent the low frequency and high frequency features of the infrared image, and the low frequency and high frequency features of the visible image, respectively. This decomposition enables modality-specific processing of spectral components and serves as the foundation for subsequent cross-attention interaction and energy-based fusion.

Considering that existing methods often neglect the complementary information between different modalities and suffer from misalignment between modal features, we adopt an extended cross-attention mechanism to perform interactive enhancement of the decomposed frequency features. Specifically, for each modality, the module not only produces an enhanced frequency feature but also outputs a corresponding attention map that highlights the most informative regions. Formally, this can be expressed as:

$$(F_{\text{low,VIS}}^{\text{enh}}, A_{\text{VIS}}) = \text{CrossAttention}(F_{\text{VIS}}^{\text{low}}, F_{\text{IR}}^{\text{high}}, F_{\text{IR}}^{\text{high}}), \quad (F_{\text{high,IR}}^{\text{enh}}, A_{\text{IR}}) = \text{CrossAttention}(F_{\text{IR}}^{\text{high}}, F_{\text{VIS}}^{\text{low}}, F_{\text{VIS}}^{\text{low}}).$$

$$(2)$$

where $F_{\text{low,VIS}}^{\text{enh}}$ and $F_{\text{high,IR}}^{\text{enh}}$ represent the enhanced visible low-frequency and infrared high-frequency features, respectively, while $A_{\text{VIS}}$ and $A_{\text{IR}}$ are the corresponding attention maps. This design allows the visible low-frequency features to incorporate high-frequency details from the infrared modality, and vice versa, facilitating cross-modal feature fusion and implicitly generating soft ROIs for subsequent energy-based weighting. For the pixel coordinates $(u, v)$ of the ROI location, the low- and high-frequency energies are defined as:

$$e_{\text{low}}(u, v) = A_{\text{VIS}}(u, v) |F_{\text{low,VIS}}^{\text{enh}}(u, v)|^2, \quad e_{\text{high}}(u, v) = A_{\text{IR}}(u, v) |F_{\text{high,IR}}^{\text{enh}}(u, v)|^2$$

We then compute the pixel-wise energy difference and predict the dynamic fusion weight:

$$\Delta e(u, v) = e_{\text{low}}(u, v) - e_{\text{high}}(u, v), \quad W(u, v) = \sigma(g_\theta(\Delta e(u, v))), \qquad W \in [0, 1]^{H \times W}, \qquad (3)$$

where $g_\theta$ is a lightweight learnable predictor (e.g., a $1 \times 1$ convolution) and $\sigma$ is the Sigmoid activation.

These pixel-wise weights $W(u, v)$ adaptively balance the contributions of low- and high-frequency information inside the key regions and guarantee smooth transitions to surrounding areas, thereby preserving the integrity of the overall structural information. For the final frequency-feature enhancement, the fused representation is generated by combining the enhanced frequency-domain features using the predicted weights, and then transforming the result back to the spatial domain through a single inverse transform:

$$F_{\text{fused}}(u, v) = W(u, v) \cdot F_{\text{low,VIS}}^{\text{enh}}(u, v) + (1 - W(u, v)) \cdot F_{\text{high,IR}}^{\text{enh}}(u, v), \quad I_{\text{fused}}(x, y) = \mathcal{F}^{-1}(F_{\text{fused}}(u, v)) \quad (4)$$

where $\mathcal{F}^{-1}(\cdot)$ denotes the inverse Fourier transform and $W \in [0, 1]^{H \times W}$ is the pixel-wise dynamic weight predicted from the ROI energy differences.

This operation adaptively balances the visible low-frequency and infrared high-frequency contributions in the frequency domain, ensuring smooth transitions across the key regions while preserving the global structural integrity. Finally, the enhanced outputs for the two modalities are obtained by adding the fused result back to their respective original spatial features:

$$x_1 = I_{\text{fused}}(x, y) + f_{\text{IR}}(x, y), \quad x_2 = I_{\text{fused}}(x, y) + f_{\text{VIS}}(x, y) \qquad (5)$$

where $x_1$ and $x_2$ represent the final infrared-enhanced and visible-light-enhanced features, respectively.

## 3.2 Activation Region Guided Fusion Enhancement

Salient regions (such as thermal targets or highlighted textures) can effectively guide feature alignment within the modality and help address spatial misalignment between modalities. However, existing methods typically extract salient regions through fixed attention mechanisms, which are insufficient for handling modality alignment during feature fusion. Therefore, we propose a dynamically guided attention mechanism that adaptively focuses on salient regions to enhance cross modal alignment.

As shown in Figure 2, the activation region guided fusion modules first inputs the features $x_1$ and $x_2$ output by the dynamic frequency domain enhancement module into the channel attention and spatial attention mechanismsHu et al. (2018). The channel attention generates the channel attention map $W_n^c \in \mathbb{R}^{2 \times 1 \times 1 \times c}$ through concatenation, global pooling, and a multilayer perceptron; the spatial attention generates the spatial attention map $W_n^s \in \mathbb{R}^{H \times W \times 1}$ through pooling, concatenation, and convolution. Finally, the obtained channel attention and spatial attention can be expressed as: $g_c = W_n^c \in \mathbb{R}^{1 \times 1 \times C}$ and the spatial attention is $g_s = W_n^s \in \mathbb{R}^{H \times W \times 1}$.

The feature activation extraction process is as follows. For the visible light feature map $x_1$ and the infrared feature map $x_2$, channel fusion is first performed: $F_c(x, y) = x_1 \otimes x_2$. Then, the maximum activation region of the spatial global information is extracted along the dimension of a single feature channel:

$$M_i^h = \max_{x=1}^{W} F_c(:, x) \in \mathbb{R}^{B \times C \times H \times 1}, \quad M_j^w = \max_{y=1}^{H} F_c(y, :) \in \mathbb{R}^{B \times C \times 1 \times W} \tag{6}$$

where $M_a, M_b \in \mathbb{R}^{B \times C \times H \times W}$ represent the activation maps of the two input modalities. They are given by $M_a = M_a^h \otimes M_b^w$ and $M_b = M_a^h \otimes M_b^w$. The following guided attention fusion process is carried out in four steps:

(1) Nonlinear feature fusion: A weighted geometric fusion strategy is adopted here to enhance the synergistic effect of the dual attention features:

$$M_{\text{fused}} = (\alpha M_a + \beta M_b) \odot \sqrt{|M_a \odot M_b|} \tag{7}$$

where $\alpha = \beta = 0.5$ is a tunable weight (default value $\alpha = \beta = 0.5$), and $\odot$ denotes element-wise multiplication. The geometric mean $\sqrt{|M_a \odot M_b|}$ strengthens the co-activated regions of the input features, aligning with the "consensus-first" principle in guided fusion.

(2) Next, local context normalization is applied to the features. A 3×3 local average pooling is used to introduce a smoothing constraint $\hat{M}_{\text{fused}} = \text{AvgPool}_{3 \times 3}(M_{\text{fused}})$, followed by normalization: mathop $\bar{M} = \text{LayerNorm}\left(\hat{M_{\text{fused}}}\right)$. This step suppresses high-frequency noise and preserves a smooth saliency distribution that aligns with human visual perception.

(3) Adaptive thresholding: The saliency threshold is dynamically determined based on image content:

$$\tau = \gamma \cdot \mathbb{E}\left[\overline{M}_{\text{fused}}\right] \tag{8}$$

where $\mathbb{E}\left[\bar{M}_{\text{fused}}\right]$ represents the mean value of all elements in the fused saliency map $\overline{M}_{\text{fused}}$ resulting in a scalar. This scalar serves as the baseline for the threshold, which is then multiplied by the scaling factor $\gamma$ (default value 0.5) to achieve adaptive thresholding. The final binary mask $M_{\text{mask}} = \mathbb{I}\left(\overline{M}_{\text{fused}} \geq \tau\right)$ is generated through threshold comparison. where the symbol $\mathbb{I}(\cdot)$ denotes the indicator function, which is used to evaluate a given condition. This adaptive mechanism ensures stable saliency detection sensitivity across different input images.

(4) Guided Fusion Application: The generated mask $M_{fused}$ can be used to guide multimodal image fusion: In regions with high mask response, spatial details (e.g., PAM features) are preferentially preserved.In regions with low response, channel features (e.g., CAM features) are emphasized.This can be formulated as:

$$F_{\text{out}} = M_{\text{mask}} \odot g_s + (1 - M_{\text{mask}}) \odot g_c \tag{9}$$

In summary, we propose an activation region guided fusion module that uses the activation region to guide the attention mechanism to focus on important cross modal salient regions or common saliency areas. This effectively guides feature alignment within the modalities and helps address the spatial misalignment issue during cross modal fusion. Finally, the output of this module is $x'_1 = F_{out} \odot x_1 + x_1$ and $x'_2 = F_{out} \odot x_2 + x_2$.

### 3.3 HIERARCHICAL SEMANTIC FEATURE ENHANCEMENT

Considering the importance of deep semantic information for downstream tasks such as object segmentation and detection, this paper addresses the issue that existing methods relying on a single Transformer or static CNN lead to incomplete extraction of deep semantic information. It proposes a dynamic semantic information mining module that integrates Transformer and CNN. This module employs dynamic masking to adjust the global and local semantic information of the fused features based on convolutional response strength, selectively retaining deep semantic information.

As shown in Figure 2, the output feature $x_1'$ from the activation region guided fusion enhancement module is used to compute the self-attention matrices: $Q^{x_1'}, K^{x_1'}, V^{x_1'} \in \mathbb{R}^{HW \times C}$. Dot-product attention is calculated to weight $V^{x_1'}$, producing the self-attention output: $\text{Attention} = \text{soft} \max\left(\frac{Q^{x_1'} K^{x_1'}{}^{T}}{\sqrt{d_k}}\right) V^{x_1'}$. Then, a feed-forward network (FFN) fuses the input with the attention output to update the features: $X_1^{\text{self}} = FFN(x_1' + \text{Attention})$. The traditional MLP in the FFN is replaced by depthwise separable and $1 \times 1$ convolutions to reduce parameters. Similarly, the self-attention output for feature $x_2'$ is denoted as $X_2^{\text{self}}$.

The calculation process of the core components of this module is described as follows: first, the input tensor $X_1^{\text{self}}$, with shape (B,N,C)—where B is the batch size and $N = H \times W$ is the spatial dimension—is reshaped into the standard convolutional feature map format(B,C,H,W), denoted as $X_1^{\text{self\_conv}} \in \mathbb{R}^{B \times C \times H \times W}$. The following describes the generation of the dynamic mask: $R = W_{\text{conv}} * X_1^{\text{self\_conv}} \in \mathbb{R}^{B \times C_{\text{out}} \times H \times W}$. where $R$ is the response map obtained by applying the convolution kernel $W_{\text{conv}} \in \mathbb{R}^{C_{\text{out}} \times C \times K \times K}$ to the feature map. Among them, $C_{\text{out}}$ represents the number of output channels of the convolutional layer, and $K$ represents the spatial size of the convolution kernel. The dynamic modulation parameters are then generated from the response map: $\gamma = \text{GlobalAvgPool}(R) \in \mathbb{R}^{C_{\text{out}}}$, $\beta = \text{GlobalMaxPool}(R) \in \mathbb{R}^{C_{\text{out}}}$, where $\gamma$ and $\beta$ represent channel-wise scaling and shifting parameters respectively, and $\theta$ is a learnable scaling factor (initialized to 0.5). The final kernel adjustment is performed as:

$$W_{\text{adjusted}}[c,:,:,:] = (\theta \cdot \gamma[c] + (1 - \theta) \cdot \beta[c]) \cdot W_{\text{conv}}[c,:,:,:], \quad \forall c \in [1, C_{\text{out}}] \tag{10}$$

This channel-wise modulation adaptively adjusts each output channel of the convolution kernel based on the feature responses. Next, a convolution is performed on the entire attention output using the mask-adjusted kernel: $out\_global_1 = \text{Conv2D}(X_1^{\text{self\_conv}}, W_{\text{adjusted}})$. Similarly, the same operation is applied to $X_2^{\text{self}}$ (after reshaping to $X_2^{\text{self\_conv}}$) to obtain $out\_global_2$.

For the local semantic information in the attention features, the following describes the feature separation process. The self-attention output $X_1^{\text{self\_conv}}$ is split along the channel dimension into $G$ groups:

$$\{feature_i\}_{i=1}^{G} = \text{Split}(X_1^{\text{self\_conv}}), \quad \text{where } feature_i \in \mathbb{R}^{B \times (C/G) \times H \times W} \tag{11}$$

For each feature group, we generate group-specific modulation parameters: $\gamma_i = \text{GlobalAvgPool}(W'_{\text{conv}} * feature_i) \in \mathbb{R}^{C_{\text{out}}/G}$ $(i = 1, ..., G)$, adjust the group convolution kernel as $W_{\text{adjusted},i}[c,:,:,:] = \gamma_i[c] \cdot W'_{\text{conv}}[c,:,:,:]$ $(\forall c \in [1, C_{\text{out}}/G])$, and compute the local feature $local_i = \text{Conv2D}(feature_i, W_{\text{adjusted},i})$, where $W'_{\text{conv}}$ is a group convolution kernel. Finally, all local features are concatenated and permuted to restore the original dimensions:

$$out\_local_1 = \text{Permute}(\text{Concat}(local_1, ..., local_G)) \in \mathbb{R}^{B \times C \times H \times W} \tag{12}$$

Similarly, for the $X_2^{\text{self}}$ features, after undergoing the same processing, the result is denoted as $out\_local_2$. To enhance the fused features in both global and local semantics, we adopt an interactive attention mechanism. The features after dynamic masked convolution $out\_global_1$ and $out\_global_2$ mutually enhance self-attention outputs, while local features $out\_local_1$ and $out\_local_2$ similarly enhance corresponding self-attention features. Taking the enhancement of $X_2^{\text{self}}$ by $out\_global_1$ as an example: $Attention_2 = \text{softmax}\left(\frac{Q^{X_2^{self}} K^{out_{g}local_1}{}^{T}}{\sqrt{d_k}}\right) V^{out_{g}local_1}$, $X_{global_2}^{cross} = \text{FFN}(X_2^{\text{self}} + Attention_2)$.

The other three cross outputs $X_{global_1}^{cross}$, $X_{local_1}^{cross}$, and $X_{local_2}^{cross}$ are computed similarly. Finally, the obtained cross-semantic features are fused using concatenation and element-wise multiplication: $x_{out} = \left(X_{global_1}^{cross} \oplus X_{global_2}^{cross}\right) \odot \left(X_{local_1}^{cross} \oplus X_{local_2}^{cross}\right)$, where $\oplus$ denotes feature concatenation operation and $\odot$ denotes element-wise multiplication. For the semantic segmentation head, we adopt

the multilayer perceptron (MLP) decoder from SegFormerXie et al. (2021) because it is simple, lightweight, and effectively captures global scene semantics. The semantic segmentation is supervised using the standard cross-entropy loss, formalized as: $\mathcal{L}_{\text{seg}} = -\sum P \log I^S$, where $P$ denotes the ground truth label, and $I^S$ represents the classification probability output by the segmentation head.

# 4 EXPERIMENTS

## 4.1 DATASETS AND IMPLEMENTATION

We evaluate on MFNet (1,569 pairs), PST900 (1,038), and FMB (1,500) with test sets of 393, 288, and 280 pairs at 480×640, 720×1280, and 600×800. The model trains 500 epochs per dataset with batch size 3, learning rate $1e-6$, using Adam on dual RTX 3090 GPUs.

## 4.2 SEMANTIC SEGMENTATION

We conducted comparative experiments on semantic segmentation by evaluating our method against nine state-of-the-art approaches: SeAFusion Tang et al. (2022), EGFNet Zhou et al. (2022), LAS-Net Li et al. (2023b), SegMiF Liu et al. (2023a), MDRNet+ Wang et al. (2023), SGFNet Zhou et al. (2023), MMSNet Liang et al. (2023), EAEFNet Liang et al. (2023), MRFSZhang et al. (2024), MultiTVIF Zhao et al. (2025), and SAGEWu et al. (2025a). In the comparative experiments, we reproduced and retrained all methods on the three datasets. As shown in Tables 1, 2, and 3, our proposed method consistently achieves superior performance across all datasets, with the most significant improvement observed on the MFNet dataset. This is primarily attributed to the advantages of MFNet in terms of spatial alignment and scene diversity between infrared and visible images.

Table 1: Semantic segmentation on the MFNet dataset.

| Method | Car | Person | Bike | Curve | Car Stop | Guar. | Cone | Bump | mIoU |
|---|---|---|---|---|---|---|---|---|---|
| SeAFusion | 84.2 | 71.1 | 58.7 | 33.1 | 20.1 | 0.0 | 40.4 | 33.9 | 48.8 |
| EGFNet | 87.6 | 69.8 | 58.8 | 42.8 | 33.8 | 7.0 | 48.3 | 47.1 | 54.8 |
| LASNet | 84.2 | 67.1 | 56.9 | 41.1 | 39.6 | 18.9 | 48.8 | 40.1 | 54.9 |
| SegMiF | 87.8 | 71.4 | 63.2 | 47.5 | 31.1 | 0.0 | 48.9 | 50.3 | 56.1 |
| MDRNet+ | 87.1 | 69.8 | 60.9 | 47.8 | 34.2 | 8.2 | 50.2 | 55.0 | 56.8 |
| SGFNet | 88.4 | 77.6 | 64.3 | 45.8 | 31.0 | 6.0 | 57.1 | 55.0 | 57.6 |
| MMNet | 89.2 | 69.1 | 63.5 | 46.4 | 41.9 | 8.8 | 48.8 | 57.6 | 58.1 |
| EAEFNet | 87.6 | 72.6 | 63.8 | 48.6 | 35.0 | 14.2 | 52.4 | 58.3 | 58.9 |
| MRFS | 89.4 | 75.4 | 65.0 | 49.0 | 37.2 | 5.4 | 53.1 | 58.8 | 59.1 |
| **MFS** | **96.6** | **80.4** | **74.0** | **65.0** | **44.2** | **21.4** | **57.1** | **65.8** | **63.8** |

Table 2: Semantic segmentation on the PST900.

| Method | Hand-Drill | BackPack | Fire-Extinguisher | Survivor | mIoU |
|---|---|---|---|---|---|
| SeAFusion | 65.6 | 59.6 | 41.1 | 29.5 | 58.9 |
| EGFNet | 64.7 | 83.1 | 71.3 | 74.3 | 78.5 |
| LASNet | 77.8 | 86.5 | 82.8 | 75.5 | 84.4 |
| MDRNet+ | 63.0 | 76.3 | 63.5 | 71.3 | 74.6 |
| SegMiF | 66.0 | 81.4 | 76.3 | 75.5 | 79.7 |
| MMNet | 62.4 | 89.2 | 73.3 | 74.7 | 79.8 |
| SGFNet | 82.8 | 75.8 | 79.9 | 72.7 | 82.1 |
| EAEFNet | 80.4 | 87.7 | 84.0 | 76.2 | 85.6 |
| MRFS | 79.4 | 87.4 | 88.0 | 79.6 | 86.9 |
| **MFS** | **81.3** | **89.5** | **90.1** | **80.5** | **88.3** |

Table 3: Semantic segmentation on the FMB.

| Method | Car | Person | Truck | T-Lamp | T-Sign | Buil. | Vege. | Pole | mIoU |
|---|---|---|---|---|---|---|---|---|---|
| SeAFusion | 76.2 | 59.6 | 15.1 | 34.4 | 68.0 | 80.1 | 83.5 | 38.4 | 51.9 |
| LASNet | 73.2 | 58.3 | 33.1 | 32.6 | 68.5 | 80.8 | 83.4 | 41.0 | 55.7 |
| SegMiF | 78.7 | 65.5 | 42.4 | 35.6 | 71.7 | 80.1 | 85.1 | 35.7 | 58.5 |
| MDRNet+ | 75.4 | 67.0 | 27.0 | 41.4 | 68.4 | 79.8 | 82.7 | 45.3 | 55.5 |
| SGFNet | 75.0 | 67.2 | 34.6 | 45.8 | 71.4 | 78.2 | 82.7 | 42.8 | 56.0 |
| EAEFNet | 79.7 | 61.6 | 22.5 | 34.3 | 74.6 | 82.3 | 86.6 | 46.2 | 58.0 |
| MRFS | 76.2 | 71.3 | 34.4 | 50.1 | 75.8 | 85.5 | 87.0 | 53.6 | 61.2 |
| MultiTVIF | 77.8 | 69.4 | 38.2 | **51.4** | 76.2 | 85.8 | 86.5 | 52.9 | 61.8 |
| SAGE | 77.2 | 72.6 | 36.2 | 48.7 | 76.1 | 83.9 | 87.4 | 51.8 | 61.5 |
| **MFS** | **81.7** | **73.3** | **39.8** | 45.7 | **76.2** | **86.1** | **88.2** | **53.7** | **62.6** |

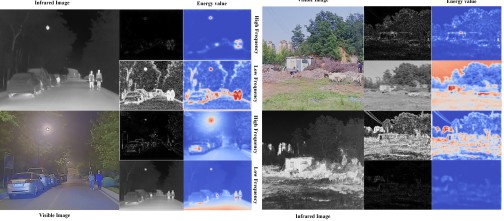

Figure 3: The correspondence between image energy and frequency

## 4.3 ABLATION EXPERIMENTS

Ablation studies validate the necessity of each component through module removal. The multimodal dynamic frequency-domain feature enhancement module (DFD) improves image details by enhancing the complementarity between frequency and energy features; the activation region-guided fusion module (ARG) focuses on salient regions in multimodal data to enrich key information in the fused image; the hierarchical semantic feature enhancement module (HSF) strengthens global and local semantic representations through attention mechanisms and dynamic convolutional masks. As shown in Table 4, the synergistic effect of these three modules achieves optimal segmentation performance on the FMB dataset, with consistent patterns observed on other datasets. Removing any module impairs feature robustness, semantic enhancement, or guided fusion capability, thereby compromising image clarity, structural integrity, and semantic completeness.

Table 4: Ablation Study on Individual Modules.

| Model | Car | Person | Truck | T-Lamp | T-Sign | Buil. | Vege. | Pole | mIoU |
|---|---|---|---|---|---|---|---|---|---|
| DFD | 80.5 | 70.1 | 38.6 | 44.1 | 75.2 | 84.9 | 87.3 | 52.8 | 61.5 |
| ARG | 77.4 | 67.7 | 36.2 | 40.8 | 72.9 | 84.9 | 86.2 | 50.9 | 59.7 |
| HSF | 78.9 | 70.0 | 38.4 | 43.7 | 74.0 | 84.0 | 86.8 | 51.5 | 60.4 |
| **MFS** | **81.7** | **73.3** | **39.8** | **45.7** | **76.2** | **86.1** | **88.2** | **53.7** | **62.6** |

Table 5: Ablation Study on ARG and HSF Key Components.

| Model | Car | Person | Truck | T-Lamp | T-Sign | Buil. | Vege. | Pole | mIoU |
|---|---|---|---|---|---|---|---|---|---|
| ARG- | 78.3 | 68.9 | 37.4 | 41.2 | 74.4 | 85.9 | 87.2 | 51.8 | 61.5 |
| HSF- | 79.7 | 71.1 | 38.8 | 43.9 | 75.6 | 84.9 | 87.3 | 52.6 | 61.2 |
| **MFS** | **81.7** | **73.3** | **39.8** | **45.7** | **76.2** | **86.1** | **88.2** | **53.7** | **62.6** |

As shown in Table 5, removing the attention fusion component from the Activation Region Guided Fusion module (ARG) significantly degrades model performance. This component identifies key regions through activation areas and allocates attention weights accordingly. Similarly, removing the dynamic convolutional mask component from the Hierarchical Semantic Feature Enhancement module (HSF) also leads to performance degradation. This component enhances cross-modal collaboration through dynamic modulation to capture deep semantic relationships.

### 4.4 VISUALIZATION RESULTS

Visualization is essential in computer vision, intuitively showing bounding boxes for object detection and pixel-level classification for semantic segmentation. The figure below presents the results on the FMB and MFNet datasets. We conducted visual comparison experiments on the semantic segmentation task to evaluate the visual performance of our method against seven state-of-the-art algorithms: EGFNet, LASNet , SegMiF , MDRNet+, SAGE, and MultiTVIF, MRFS. As shown in Figure 4, the experimental results demonstrate that our method achieves superior visual segmentation performance, characterized by the best classification accuracy and complete object contour delineation. For instance, our approach effectively preserves fine-grained details in the contours of pedestrians and vehicles, presenting vivid shapes, whereas other methods can only identify rough regions.

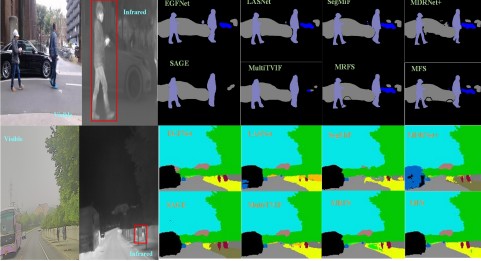
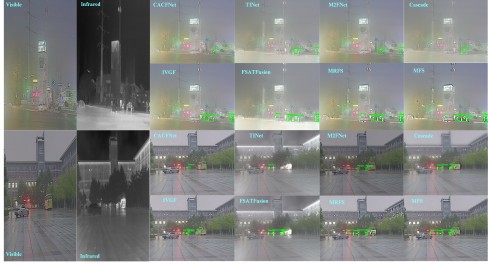

Figure 4: Segmentation Results on the MFNet and FMB      Figure 5: Detection Results on the MFNet and FMB

We conducted comprehensive experimental evaluations for object detection tasks, comparing our method with seven state-of-the-art approaches: CACFNet Zhou et al. (2024), TINet Zhang et al. (2023), M²FNet Liu et al. (2024b), Cascade Li et al. (2024b), IVGF Liu et al. (2024a), and FSATFusion Zhang et al. (2025a), MRFSZhang et al. (2024). The experimental procedure involved first fusing infrared and visible-light images into more information-rich representations using each respective model, then feeding the fused images into the YOLOv5Jocher (2020) detector to evaluate detection performance. As demonstrated in Figures 5, the results show that our method achieves superior performance in object detection, characterized by higher localization accuracy and more complete bounding box regression. Specifically, our approach precisely captures human poses and detects faint targets (e.g., infants), while competing methods suffer from missed detections or bounding box misalignment.

### 4.5 GAINS FROM THE ACTIVATION REGION GUIDED FUSION

To evaluate the effectiveness of the Activation Region Guided Fusion Enhancement module (ARG) and the Hierarchical Semantic Feature Enhancement module (HSF) in improving multimodal information fusion and semantic information learning, we respectively integrate these two modules into the Tufusion Zhao et al. (2024b) and MATCNN Liu et al. (2025) frameworks for comparative validation. In the experiments, we perform quantitative analysis on the TNO dataset Toet (2017) using the following five key evaluation metrics: Mutual Information (MI), which measures the dependency between the fused and source images; Entropy (EN), reflecting the information richness

of the fused result; Standard Deviation (SD), indicating contrast quality; the Edge and Texture Detection Metrics (Qabf), evaluating edge preservation; and Spatial Frequency (SF), assessing spatial detail activity. As shown in Tables 6 and 7, the experimental results demonstrate that the ARG and HSF modules effectively help preserve key information in the fused images and enhance semantic information.

Table 6: Enhance feature fusion through the ARG module.

| Method | MI | EN | SD | Qabf | SF |
|---|---|---|---|---|---|
| Tufusion | 2.3796 | 6.5051 | 0.1143 | 0.2515 | 0.0218 |
| **Tufusion+** | **2.4149** | **6.8397** | **0.1368** | **0.3689** | **0.0285** |
| MATCNN | 3.3978 | 6.9862 | 0.1913 | **0.5291** | 0.05015 |
| **MATCNN+** | **3.4234** | **7.0482** | **0.1945** | 0.5189 | **0.05258** |

Table 7: Feature fusion enhancement via HSF module.

| Method | MI | EN | SD | Qabf | SF |
|---|---|---|---|---|---|
| Tufusion | 2.8444 | 6.4944 | 0.1587 | 0.1852 | 0.02443 |
| **Tufusion+** | **3.0492** | **6.6237** | 0.1368 | **0.2347** | **0.02558** |
| MATCNN | 4.7847 | 6.7987 | 0.1904 | 0.5983 | 0.04815 |
| **MATCNN+** | **4.9695** | **7.1437** | **0.1945** | 0.5659 | **0.05167** |

## 4.6 GAIN FROM THE HIERARCHICAL SEMANTIC FEATURE

To verify the effectiveness of the Hierarchical Semantic Feature Enhancement (HSF) module in semantic modeling and the Activation Region Guided (ARG) fusion module in key region exploration, we integrate them separately into the Mask DINO Li et al. (2023c) and DI-MaskDINO Xu et al. (2024) frameworks for comparative experiments. Eight key metrics are used: $AP^{box}$, $AP_S^{box}$, $AP_M^{box}$, $AP_L^{box}$, $AP^{mask}$, $AP_S^{mask}$, $AP_M^{mask}$, and $AP_L^{mask}$. Object detection and semantic segmentation experiments are conducted on the COCO Lin et al. (2014) dataset. As shown in Tables 8 and 9, the experimental results demonstrate that these modules significantly improve the performance of the original models, validating their effectiveness in enhancing semantic information and capturing key image regions, thereby improving the model's ability to understand and recognize targets.

Table 8: Semantic information enhancement of features based on the HSF module.

| Method | Epochs | $AP^{box}$ | $AP_S^{box}$ | $AP_M^{box}$ | $AP_L^{box}$ |
|---|---|---|---|---|---|
| MaskDINO | 12 | 52.2 | 34.8 | 55.6 | **69.9** |
| **MaskDINO+** | 12 | **53.1** | **36.2** | **56.1** | 69.2 |
| DI-MaskDINO | 12 | 53.3 | 36.7 | **56.7** | 70.4 |
| **DI-MaskDINO+** | 12 | **53.8** | **37.5** | 56.4 | **71.5** |
| MaskDINO | 50 | 56.8 | 40.2 | 60.2 | **72.3** |
| **MaskDINO+** | 50 | **57.2** | **40.8** | **60.4** | 72.2 |
| DI-MaskDINO | 50 | 57.8 | 41.5 | **61.2** | 73.9 |
| **DI-MaskDINO+** | 50 | **58.7** | **42.7** | 60.6 | **74.5** |

Table 9: Semantic information enhancement of features based on the HSF module.

| Method | Epochs | $AP^{mask}$ | $AP_S^{mask}$ | $AP_M^{mask}$ | $AP_L^{mask}$ |
|---|---|---|---|---|---|
| MaskDINO | 12 | 47.2 | 26.3 | 50.3 | 69.1 |
| **MaskDINO+** | 12 | **48.0** | **26.9** | 50.0 | **69.9** |
| DI-MaskDINO | 12 | 47.9 | 27.7 | 51.5 | 69.3 |
| **DI-MaskDINO+** | 12 | **48.8** | **28.9** | **52.4** | **70.6** |
| MaskDINO | 50 | 51.0 | 31.3 | 54.1 | 71.2 |
| **MaskDINO+** | 50 | **51.4** | **31.7** | **54.5** | **72.0** |
| DI-MaskDINO | 50 | 51.8 | 31.8 | 55.1 | 72.2 |
| **DI-MaskDINO+** | 50 | **52.6** | **32.5** | **56.3** | **72.8** |

We visualized the different training stages (i.e., epoch = 12 and epoch = 50) of MaskDINO and DI-MaskDINO on the COCO dataset. As shown in Figure 6, the effectiveness of the proposed HSF module is clearly demonstrated. Our method exhibits higher robustness in detection tasks, especially for small and medium-sized objects. Similarly, Figure 7 illustrates the effectiveness of the proposed ARG module in enhancing semantic features and reinforcing key target information.

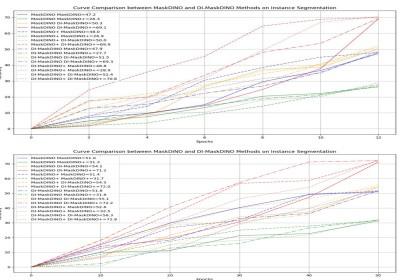 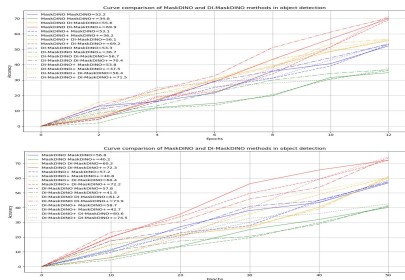

Figure 6: Segmentation performance curves on the COCO dataset    Figure 7: Detection performance curves on the COCO dataset

## 4.7 CONCLUSION

This paper proposes a multimodal semantic segmentation framework combining frequency-domain dynamic routing and activation region guidance. By leveraging edge enhancement, hybrid attention, and deep semantic learning modules, it achieves efficient image fusion, segmentation, and object detection. Experiments show strong robustness and generalization, especially for small, weak, and occluded targets.

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

# A APPENDIX

## A.1 VISUALIZATION

We further conducted a comprehensive visual evaluation experiment on visible and infrared image fusion tasks, comparing our method with seven state-of-the-art approaches, including SeAFusion Li et al. (2022), DATFuseTang et al. (2023), Gan-HALu et al. (2024), ADF-NetShen et al. (2024), U2FusionXu et al. (2022), TGFuseRao et al. (2023a), and CDDFuse Zhao et al. (2023), where all experiments were performed under identical hardware configurations to ensure fair and consistent visual comparisons. As demonstrated in Figures 8, the experimental results reveal that our method exhibits remarkable advantages in both multi-modal feature preservation and detail enhancement: specifically, it excels at retaining fine visible-light textures (such as road signs and building contours) where other methods tend to produce blurred or incomplete results; it significantly enhances thermal radiation targets (like pedestrians and vehicles) by presenting clearer thermal signatures without overexposure or low-contrast issues; and most importantly, it achieves an optimal balance between natural visual appearance and target saliency that outperforms all competing methods. These experimental findings collectively confirm that our method has reached state-of-the-art performance in visible-infrared image fusion tasks.

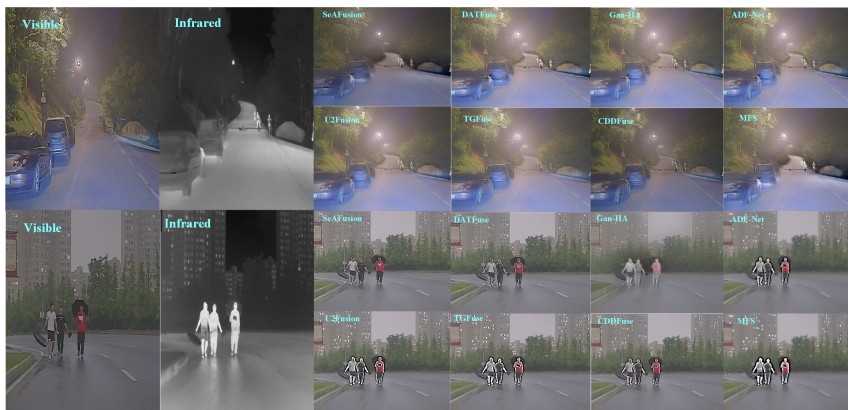

Figure 8: Qualitative Fusion Results on the MFNet and FMB Dataset

## A.2 ABLATION EXPERIMENTS

Ablation studies effectively validate the necessity of each module within the model. By removing or replacing key components in these modules, the individual contributions of each part are clearly demonstrated, which enhances the credibility of the overall conclusions. Based on this, we conducted extensive ablation experiments to evaluate the modular design of the proposed method. Figures 9 and 10 present the visual ablation analysis results for the three key modules of our model. Each module plays an important role in both semantic image segmentation and object detection. The multimodal dynamic frequency domain feature enhancement module (DFD) strengthens complementary information between modalities in the frequency and energy domains, improving detail clarity and structural reconstruction capability and facilitating the extraction of rich feature information. The activation region guided multimodal fusion module (ARG) uses activation regions to guide

the attention mechanism to focus on key areas in the image, thereby enhancing the accurate fusion of targets. The hierarchical semantic feature enhancement module (HSF) dynamically models deep semantic information through masks based on both global and local regions, improving the model's understanding of multi-source semantic information.

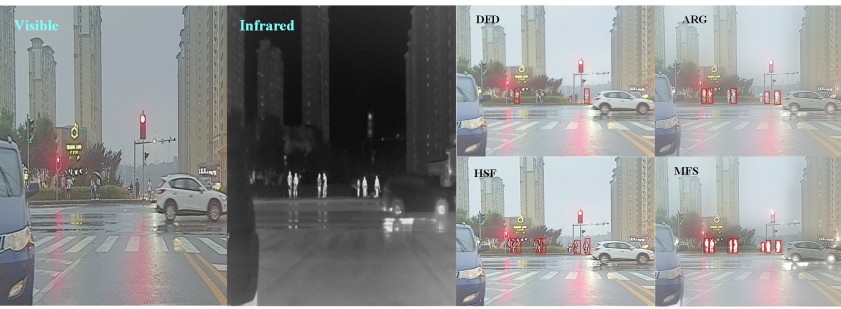

Figure 9: Ablation study on Object Detection based on visible and infrared image fusion.

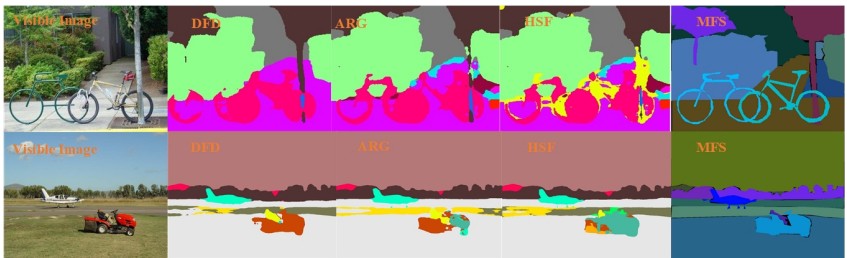

Figure 10: Ablation study on Semantic Segmentation based on visible and infrared image fusion

In the visible and infrared image fusion task, we propose an activation region guided attention fusion module (ARG). This module uses activation regions to guide the dynamic allocation of attention weights, effectively leveraging the detailed texture information of visible images and the thermal radiation information of infrared images, enabling the model to selectively focus on salient features from both modalities. As shown in Figure 11, the first row displays feature maps generated by the conventional attention mechanism, while the second row shows outputs enhanced by our guided attention. Visual comparison clearly demonstrates that our method successfully guides the model to focus on key regions of the modalities (such as structural edges and thermal targets), while effectively suppressing noise interference, thereby validating the effectiveness of the module in improving the quality of multimodal image fusion.

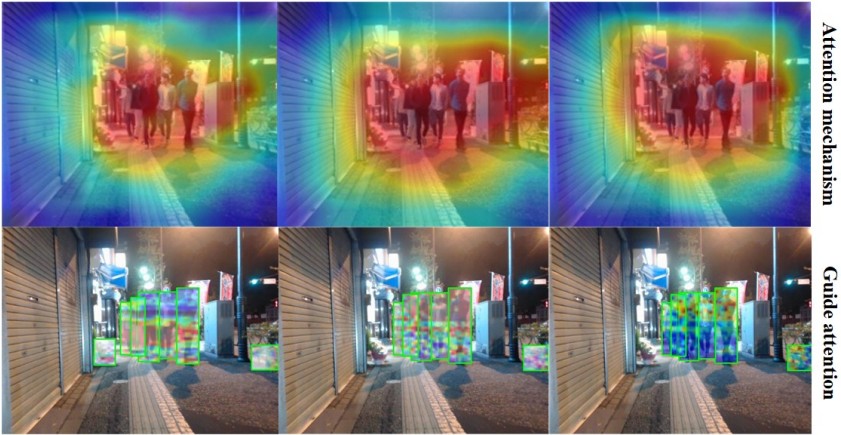

Figure 11: Visual Comparison of ARG Module Guided Attention Mechanism on the MFNet Dataset

## A.3 COMPLEXITY ANALYSIS

To comprehensively evaluate the computational complexity of different semantic segmentation and image fusion methods, we conducted a quantitative analysis of parameter count and FLOPs (see Table 10Li et al. (2023a); Wang et al. (2023); Zhou et al. (2023); Liang et al. (2023); Chi et al. (2024); Rao et al. (2023b); Zhang et al. (2024); Wu et al. (2025a); Zhao et al. (2025). As shown in Table 10, the table clearly compares the parameter count and FLOPs of each method, and our approach shows clear advantages over some advanced semantic segmentation or image fusion methods.

Table 10: Comparison of different methods in segmentation, image fusion, computational cost (FLOPs), and parameters.

| Method | Segmentation | Image Fusion | FLOPs (G) | Params (M) |
|---|---|---|---|---|
| LASNet | ✓ | ✗ | 371.03 | 93.58 |
| MDRNet+ | ✓ | ✗ | 891.82 | 210.87 |
| SGFNet | ✓ | ✗ | 225.63 | 125.12 |
| EAEFNet | ✓ | ✗ | 316.49 | 147.21 |
| LMDFusion | ✗ | ✓ | 26.67 | 44.28 |
| TGFuse | ✗ | ✓ | 137.34 | 19.34 |
| MRFS | ✓ | ✓ | 219.16 | 134.97 |
| SAGE | ✓ | ✓ | 102.53 | 13.06 |
| MultiTVIF | ✓ | ✓ | 125.21 | 2.47 |
| **MFS** | ✓ | ✓ | **11.80** | **0.34** |

## B REPRODUCIBILITY STATEMENT

The partially anonymized code of this paper is as follows: `https://anonymous.4open.science/r/MFS_Net-CB23`. I hereby commit that, if this paper is accepted, all code will be immediately open-sourced to facilitate reproducibility.

