# OpenReview forum: "MFS: A Saliency Driven Interactive Multimodal Fusion Framework for Robust Semantic Segmentation in Complex and Occluded Scenes"
_ICLR.cc/2026/Conference — ICLR 2026 Conference Desk Rejected Submission_

### Official Review · Reviewer_G12N · 2025-10-23

**Soundness:** 3
**Presentation:** 1
**Contribution:** 2
**Rating:** 2
**Confidence:** 5

**Summary:**

The paper proposes a saliency-driven multimodal fusion framework for robust semantic segmentation in complex and occluded scenes. It integrates three modules, a dynamic frequency-domain feature enhancement module that adaptively fuses infrared and visible features based on frequency energy, an activation region guided fusion enhancement module that aligns salient cross-modal regions, and a hierarchical semantic feature enhancement module combining Transformer-based global reasoning and dynamic masked convolutions for fine-grained semantics. Experiments on MFNet, PST900, and FMB datasets show that MFS consistently outperforms prior methods in segmentation accuracy.

**Strengths:**

1. The idea of combining saliency object segmentation with multimodal fusion is meaningful and is predictable to bring help.
2. The experimental results demonstrate the performance advantage of this method in the segmentation tasks of three datasets.
3. The description of this method for the designed module is relatively comprehensive, which to some extent can help achieve better reproducibility.

**Weaknesses:**

1. The formulas, tables, and figures in the paper are too compact and the font is small. Moreover, the current version of the paper is difficult to read in terms of writing, further optimization is still required in the writing.
2. Results are confined to RGB-IR segmentation, it remains unclear whether the proposed design generalizes to other multimodal combinations or real-time settings.
3. Although the saliency concept is emphasized, the method does not provide an enough explanation of how saliency improves multimodal alignment.
4. The components (such as frequency fusion) mainly extend existing paradigms rather than introducing a fundamentally new multimodal principle.

**Questions:**

1. How is the computational cost of each module? The table in the appendix shows that the overall computational cost is manageable, but as a three-stage method, an analysis of the computational cost for each module would be more persuasive.
2. Have the authors tested robustness under modality degradation (e.g., missing IR channel or noisy visible input)?

---

### Official Review · Reviewer_a3A1 · 2025-10-25

**Soundness:** 2
**Presentation:** 1
**Contribution:** 1
**Rating:** 2
**Confidence:** 3

**Summary:**

This paper proposes an interactive multimodal semantic segmentation framework based on frequency domain dynamic routing and activation region guidance. The authors claim that it is capable of addressing multimodal semantic segmentation in complex environments and outperforms existing methods.

**Strengths:**

1. This work integrates multiple core modules, including an edge feature enhancement module, an activation region guided hybrid attention module, and a deep semantic enhancement learning module, to improve multimodal segmentation performance in complex scenes.

2. The authors demonstrate the effectiveness of the proposed MFS through qualitative and quantitative experiments.

**Weaknesses:**

1. Overall, this work appears more like a collection of existing works/modules, as the utilization of frequency domain features [1-2] and hybrid attention[3] has already been extensively discussed in existing works.

[1] FMTrack: Frequency-aware Interaction and Multi-Expert Fusion for RGB-T Tracking

[2] MCFusion: Frequency Domain Characteristics Enhancement and Feature Compensation Fusion Network for RGB-T Object Detection

[3] Hybrid Attention for Robust RGB-T Pedestrian Detection in Real-World Conditions

2. The paper does not directly analyze the role of the proposed modules in complex and occluded scenarios, and it lacks new insights into this field.

3. The paper lacks analysis of efficiency, and the redundant module design may sacrifice efficiency.

4. The paper introduces numerous variables and formulas, making the methodology section difficult to understand. Additionally, inconsistent notation, such as W and H, severely affects the paper's readability.

5. The paper introduces multiple hyperparameters, such as the dynamic threshold, but lacks ablation experiments analyzing the selection of these hyperparameter values

**Questions:**

1. What is the baseline for the current method? The authors should analyze the performance improvement of the proposed modules over the baseline.

2. The paper's experiments focus on road scenes. How is MFS's generalization capability? How does it perform in other complex scenarios?

---

### Official Review · Reviewer_TkAS · 2025-10-27

**Soundness:** 2
**Presentation:** 3
**Contribution:** 3
**Rating:** 6
**Confidence:** 3

**Summary:**

This paper proposes a novel interactive multimodal semantic segmentation framework named MFS, designed to address the challenges of detecting small, weak, and occluded targets in complex scenes. The framework consists of three core modules: a dynamic frequency-domain feature enhancement module (DFD), an activation region-guided fusion module (ARG), and a hierarchical semantic feature enhancement module (HSF). By leveraging frequency-domain energy routing, activation region-guided attention, and dynamic mask-based semantic learning, the method significantly improves the quality of infrared and visible image fusion and the robustness of semantic segmentation. Comprehensive experiments on multiple public datasets demonstrate its superior performance over existing methods, particularly in object detection and semantic segmentation tasks.

**Strengths:**

1. A unified framework integrating frequency-domain analysis, attention mechanisms, and semantic learning is proposed, which is conceptually appealing.
2. Extensive experiments on multiple tasks and datasets demonstrate the competitiveness of the overall performance.
3. The code is promised to be open-sourced, which benefits reproducibility.

**Weaknesses:**

1. The design of the ablation experiments is not very reasonable. The ablation experiment baseline MFS of the paper is a complete model that includes three modules. When one module is removed (e.g., DFD-), it is actually comparing the complete model with the incomplete model.
2. The comparison with the latest SOTA methods is somewhat insufficient; there are only 2 comparison methods from 2025, while the others are from 2024 and 2023.

**Questions:**

1. Regarding the design of ablation experiments, why is the ablation result not conducted by sequentially adding the DFD, ARG, and HSF modules, but instead by removing modules?
2. Apart from mIoU, could you provide other more detailed metrics to demonstrate the model's effectiveness?

---

### Official Review · Reviewer_WWrB · 2025-10-30

**Soundness:** 3
**Presentation:** 1
**Contribution:** 2
**Rating:** 2
**Confidence:** 3

**Summary:**

This paper proposes a framework of multimodal fusion segmentation that enhances semantic segmentation robustness through saliency-driven interactive fusion in both spatial and frequency domains. Specifically, the authors propose three main modules, Dynamic Frequency-Domain Feature Enhancement (DFD), Activation Region Guided Fusion (ARG), Hierarchical Semantic Feature Enhancement (HSF), focusing on dynamically routing based on frequency-domain energy, cross model alignment enhanced by salient regions via dynamically guided attention mechanism, and integrating transformer-based global self-attention with dynamic convolutional masking, respectively. Experimental results on three semantic segmentation datasets demonstrates the outperformance of the proposed method.

**Strengths:**

1. The proposed three modules—DFD, ARG, and HSF—are technically sound and well-motivated. Each module effectively addresses a specific limitation of prior multimodal fusion models, contributing to a cohesive and robust overall framework.

2. The experimental results on semantic segmentation demonstrate the superiority of the proposed method over state-of-the-art approaches. The consistent performance gains across multiple datasets highlight the method’s strong generalization ability and practical effectiveness.

3. The ablation studies presented in Tables 4 and 5 further validate the effectiveness of each module, showing that every component contributes meaningfully to the overall performance. Additionally, the analyses in Tables 6–9 demonstrate the significant feature fusion enhancement achieved by integrating the ARG and HSF modules into existing fusion frameworks, underscoring their adaptability and utility.

**Weaknesses:**

1. The presentation quality requires significant improvement. Figures 2, 6, and 7 are of very low resolution, with text and visual elements too small to be legible, making it difficult for readers to extract meaningful information. Moreover, the method section focuses heavily on technical details without providing sufficient illustrative figures or conceptual diagrams to convey the underlying intuition. This lack of visual and conceptual clarity substantially hinders the readability and overall accessibility of the paper.

2. The paper lacks sufficient theoretical analysis to support certain design choices. In particular, the rationale behind the dynamic frequency routing predictor is not thoroughly justified. The authors should provide deeper theoretical insights or empirical evidence to explain why frequency energy serves as the optimal criterion for feature fusion. Without such discussion, the motivation for this key component remains somewhat unclear.

3. The experimental comparisons lack consistency across tables. Specifically, the results for SAGE and MultiTVIF are reported only in Table 3, but are missing from Tables 1 and 2. For a fair and comprehensive evaluation, it would be important to maintain consistency in the inclusion of baseline methods across all datasets. This would allow for a more reliable and transparent comparison of performance.

4. The paper provides extensive qualitative visualizations for object detection tasks but lacks corresponding quantitative evaluations. Without numerical comparisons against state-of-the-art methods, it is difficult to objectively assess the proposed framework’s performance and effectiveness in object detection. Including quantitative results would significantly strengthen the empirical validation and support the claimed advantages.

**Questions:**

1. Some theoretical analysis of frequency routing are recommended to explain why frequency energy serves as the optimal criterion for feature fusion.

2. I suggest the authors provide SAGE and MultiTVIF's results on MFNet and PST900 datasets, and quantitative results of the comparison on object detection task.

---

### Note · Program_Chairs · 2026-01-17
**Submission Desk Rejected by Program Chairs**

The following references in this submission do not refer to real documents and/or have major errors in bibliographic information:

 H. Li, X. J. Wu, and T. Durrani. SeAFusion: A Seasonal-Adaptive Infrared and Visible Image Fusion Network. IEEE Transactions on Multimedia, 24:1686-1697, 2022. doi: 10.1109/TMM. 2021.3076246.